# A water-soluble copolymer for storage and electron conversion in photocatalytic on-demand hydrogen evolution

Marco Hartkorn [1], Robin Kampes[2,3], Felix Müller[1], Linda Zedler [4,5], Akuila Edwards[4], Philip Rohland[2,3,4], Alexander K. Mengele [1], Stefan Zechel[2,3], Martin D. Hager[2,3,6,7], Benjamin Dietzek-Ivanšić[4,5,9], Michael Schmitt [4], Jürgen Popp [4,5], Ulrich S. Schubert [2,3,6,7,8] ✉ & Sven Rau [1] ✉

Cost- and energy-efficient long-term storage of excess solar energy remains a major bottleneck in the transition to a sustainable society. Here, we present a water-soluble redox-active copolymer containing viologen moieties that can be charged with electrons upon visible light irradiation using a tris[4,4'-bis(*tert*-butyl)−2,2'-bipyridine]ruthenium(II) complex as chromophore. In the presence of a sacrificial donor, the system achieves charging efficiencies above 80% and fully maintains this state for several days. Subsequent acidification and the addition of various catalysts enable on-demand usage of the stored electrons for proton reduction to hydrogen with up to 72% efficiency. The system further demonstrates reversibility via a simple pH switch, allowing multiple charging, storage, and catalysis cycles without time-consuming polymer isolation. The present study presents a direct on-demand hydrogen evolution method through discharging of a water-soluble polymer that functions as a temporary energy and electron storage material.

Although the amount of solar energy reaching the earth is significantly exceeding human needs[1], seasonal, regional, and diurnal fluctuations are main reasons why large-scale harvesting of this immense energy source is so far not accessible. In order to facilitate the transition from the current fossil fuel-based economy into a sustainable one, appropriate materials for storing renewable energy have to be developed[2]. Besides the use of various types of batteries[3], other materials such as inorganic semiconductors[4], molecular metal oxides[5–7], polymeric heptazine-based networks[8], metal-organic frameworks[9], molecular transition metal complexes[10], or boranes[11] to name a few, have already been employed for the storage of solar energy. In many cases, these materials enable subsequent release of the stored solar energy in the

form of hydrogen upon addition of a suitable catalyst. The catalyst's role is to either facilitate the release of bound hydrogen from storage materials[2,11,12] or it allows hydrogen evolution by combining stored electrons and protons[6,8,9]. This so-called on-demand hydrogen evolution could be of great importance to different energy-intensive industrially relevant processes such as steel production, which will in future heavily rely on a constant supply of green hydrogen[13]. With aspects such as scalability in mind, one example for alternative storage materials are redox-active polymers, which have so far been applied in organic radical batteries[14] or redox-flow batteries[15].

We could already show that the covalent attachment of a ruthenium-based photosensitizer (PS) to a polyoxometalate (POM) as

[1]Institute of Inorganic Chemistry I, Materials and Catalysis, Ulm University, Ulm, Germany. [2]Laboratory of Organic and Macromolecular Chemistry (IOMC), Friedrich Schiller University Jena, Jena, Germany. [3]Jena Center for Soft Matter (JCSM), Friedrich Schiller University Jena, Jena, Germany. [4]Institute of Physical Chemistry (IPC), Abbe Center of Photonics (ACP) Friedrich-Schiller-University Jena, Jena, Germany. [5]Leibniz Institute of Photonic Technology (IPHT) Jena, Jena, Germany. [6]Center for Energy and Environmental Chemistry Jena (CEEC Jena), Friedrich Schiller University Jena, Jena, Germany. [7]Helmholtz Institute for Polymers in Energy Applications Jena (HIPOLE Jena), Jena, Germany. [8]Helmholtz Zentrum für Materialien und Energie Berlin (HZB), Berlin, Germany. [9]Present address: Leibniz Institute of Surface Engineering (IOM), Leipzig, Germany. ✉e-mail: ulrich.schubert@uni-jena.de; sven.rau@uni-ulm.de

a storage site provides a molecular system, capable of on-demand hydrogen generation[7]. As the maximum turnover number (TON) of the Ru-POM system was limited to 1 by design, similar as for related systems[10], we wondered whether a more efficient on-demand evolution of hydrogen might be possible using the redox equivalents stored on a more stable redox-active polymer. In addition to time-delayed fuel generation, the interposition of polymeric electron storage materials offers the possibility of circumventing nonproductive light-induced processes of fully molecular systems, such as light-induced intramolecular charge redistributions during and after the formation of reductively activated catalyst sites that negatively affect the system's activity[16–18].

Building on the well-known interplay of photoredox-active transition metal complexes, methyl viologen, and different hydrogen evolving catalysts (HECs)[19–21], we aimed to couple prototype photoactive Ru(II) tris(bipyridine) complexes with viologen containing copolymers to store reducing equivalents at the polymer. This stored solar energy is finally meant to be used for on-demand hydrogen evolution, starting by the addition of an appropriate HEC and/or switching the solution to acidic conditions. Based on the reversible nature of the polymer-localized redox reaction, multiple charging and hydrogen evolution catalysis sequences were achieved (see Fig. 1).

## Results and discussion

### Light-driven charging of the polymer

The redox-active polymer, which was previously used in aqueous polymer redox-flow batteries, was synthesized according to literature procedures[22]. In short, the polymer was obtained by a radical copolymerization of a styrene derivative featuring an ammonium group in order to ensure water-solubility as well as a styrene bearing the redox-active unit—the viologen. The latter monomer was employed enabling the electron storage functionality. The polymerization was performed in water. The ratio of those two moieties in the final copolymer was 1:1, which was confirmed by NMR spectroscopy, similar to a previous study[22].

We started our investigations on the polymer-based on-demand hydrogen evolution by analyzing the first step of the process, *i.e.*, light-driven reduction of the viologen moieties. Therefore, we determined the amount of viologen units that could be reduced utilizing [Ru(tbbpy)₃]Cl₂ (tbbpy = 4,4′-di-*tert*-butyl−2,2′-bipyridine) as prototype PS and triethylamine (TEA) as sacrificial donor. The system was buffered with NaH₂PO₄ to guarantee structural integrity of the polymer, as it is able to undergo irreversible dealkylation processes at higher pH-values (see Supplementary Fig. 1)[23,24]. As shown in Fig. 2A, light-driven reduction of the copolymer is associated with the rise of two broad absorption bands centered at 528 and 872 nm. These bands are indicative of the formation of reduced viologen units as has previously been shown for structurally similar viologen-containing polymers and suitable molecular reference compounds in literature[25–28]. The broad and rather unstructured UV/vis profile deviates from the finding for molecular methyl viologen (MV²⁺), which exhibits a more fine-structured absorption profile under otherwise identical conditions[25,26]. This difference is likely associated with the higher local concentration of the viologen units in the polymer leading to a dimerization of the one-fold reduced viologens, which stabilizes their radical cationic state, represented by the absorption band at 872 nm[25,26,29,30]. After 60 min of irradiation, the slope of the rise of the absorbance at 872 nm is already substantially flattened, manifesting a plateau in the second hour of irradiation (see Fig. 2A). Having proven that photochemical reduction of the redox-active polymer is possible, we devised experiments to quantify this charging process.

In order to evaluate the efficiency of this charging process, the absorption of the fully reduced polymer was recorded using chemical reductants. Utilizing this data allows to establish a reference system, allowing a quantitative comparison of the photochemically reduced

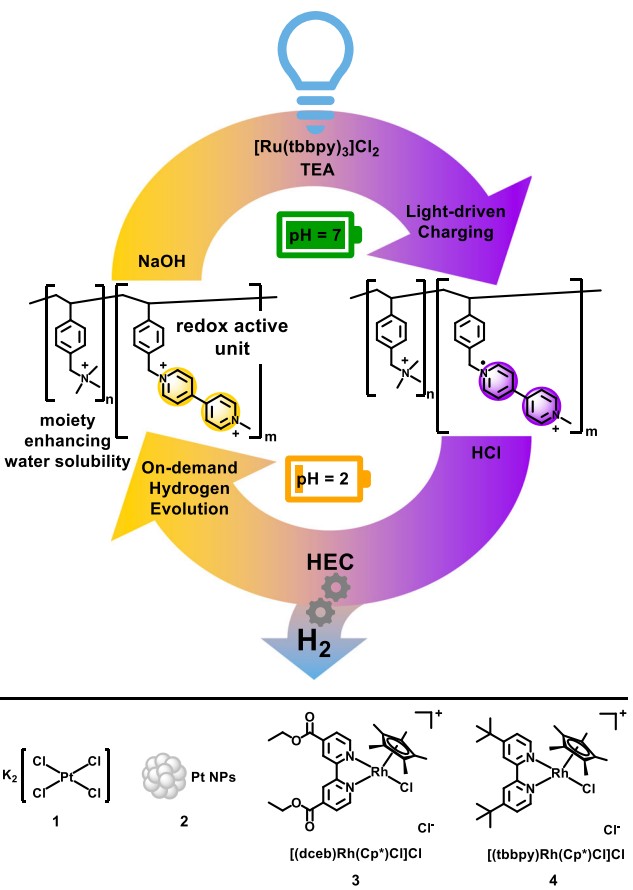

**Fig. 1 | Graphical summary of the full catalytic cycle.** Top: Graphical representation of the repeated on-demand hydrogen evolution using a viologen containing copolymer as electron storage material. Bottom: Hydrogen evolving catalysts investigated in this study (NP nanoparticle, dceb 4,4´-di(carboxyethyl) −2,2´-bipyridine, tbbpy 4,4´-di-*tert*-butyl-2,2′-bipyridine, Cp* pentamethylcyclo-pentadienyl, TEA triethylamine, HEC hydrogen evolving catalyst).

samples to the chemically reduced standard. The utilized reducing agent Na₂S₂O₄ is capable of fully reducing the redox-active viologen units at various polymer concentrations[31] (see SI for a detailed description of the experimental procedure). The resulting regression (depicted in Supplementary Fig. 2) enabled the determination of the absorption of the fully reduced polymer. Consequently, this allowed the calculation of the state of charging (SOC) of the polymer during both half reactions at any time. Accordingly, these factors, based on the respective concentration of the sample, obtained from the regression are utilized to normalize the absorption spectra presented in this work. Therefore, the cross-section at the peak maximum at 528 nm directly yields the SOC given in Fig. 2B, which shows an asymptotic convergence to about 82 ± 2% (TON = 14.8 with respect to utilized [Ru(tbbpy)₃]Cl₂) of reduced viologen units, which is reached after 2 h of irradiation. This charge is lost instantly upon aerating the sample, highlighting the necessity of the inert handling of samples (see Supplementary Fig. 3).

Addressing the incomplete charging, it can be noted that the arising absorption of the charged polymer overlaps with the absorption of the PS. A closer look at the charging trajectory, compared to the percentage that the ruthenium absorption constitutes at the irradiation wavelength, reveals that co-absorption of PS and polymer is not responsible for the incomplete trend of the SOC (see Supplementary Fig. 4). Based on emission quenching studies, the photocatalytic mechanism dominantly proceeds via a oxidative quenching pathway

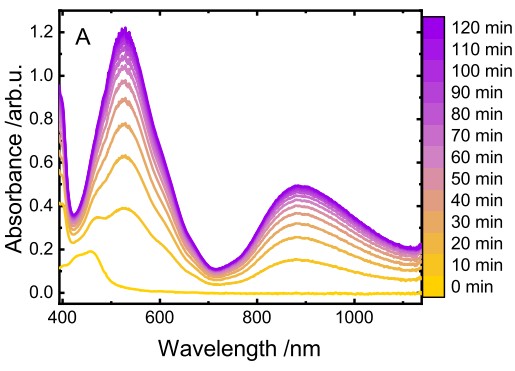
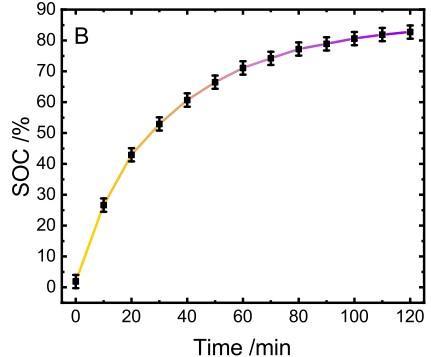

**Fig. 2 | Charging process of the polymer.** UV/vis absorption spectra during photocatalytic charging of the copolymer (225 μM with respect to methyl viologen monomer content) using [Ru(tbbpy)$_3$]Cl$_2$ (12.5 μM) in water containing 0.09 M TEA and 0.075 M NaH$_2$PO$_4$ (**A**) and the respective time course of charging (**B**). SOC refers to state of charging; mean values are displayed and error bars represent standard deviation of $n = 3$ independent measurements. Source data are provided as a Source data file.

(see Supplementary Fig. 5)[32]. The reason why no higher degree of charging was obtained is likely associated with the fact that the electrons stored on the polymer are efficiently competing with TEA for reductive quenching of the Ru chromophore (see Supplementary Fig. 5). In fact, the reduced polymer exhibits a much larger driving force for reducing the Ru complex as the reduced polymer is oxidized at −0.7 V vs. Fc$^+$/Fc whereas TEA is oxidized only at 0.3 V vs. Fc$^+$/Fc[22,33]. With increasing state of charge of the viologen copolymer, the electrochemical equilibrium potential will also be more negative, which could hamper the charging by the PS.

## Catalysis-driven discharging of the polymer

Having established the charging of the viologen units within the polymer up to 82%, the discharging of the photochemically reduced polymer utilizing various catalysts was investigated. As HECs, K$_2$PtCl$_4$ (**1**), platinum nanoparticles (Pt-NPs, **2**) and two Rh(III) complexes depicted in Fig. 1, namely [(dceb)Rh(Cp*)Cl]Cl (**3**) and [(tbbpy)Rh(Cp*)Cl]Cl (**4**) were chosen as these compounds have shown catalytic activity for hydrogen evolution in similar systems[20,34,35]. The molecular HECs differ in the bipyridine-type ligand which determines the redox potential of the rhodium center (**3** = −0.9 V vs. Fc$^+$/Fc[36,37], **4** = −1.1 V vs. Fc$^+$/Fc[37,38]), thus allowing for different driving forces of electron transfer between the reduced polymer and the putative catalytic center. The discharging of the polymer for hydrogen evolution initiated by the various HECs was followed by UV/vis spectroscopy.

Surprisingly, when only adding the HEC (20 mol% with respect to the amount of 450 μM MV$^{2+}$ units present in the buffered polymer solution, see SI for a detailed experimental description), just K$_2$PtCl$_4$ (**1**) led to a rapid discoloration of the violet solution, which is reflected by loss of the broad reduced polymer-associated absorption bands, accompanied by the formation of a black precipitate (see Supplementary Fig. 6A). These findings can be linked to the reduction of Pt(II) to Pt(0) followed by aggregation of the metal particles[39,40]. No hydrogen formation could be detected under these conditions for any of the employed HECs via gas chromatography. Further irradiation of the discolored sample led again to a recharging of the polymer proving that the principal functionality of the redox-active polymer is not affected by the discharging process and the formation of the metal particles (Supplementary Fig. 7). For **2** and **3**, the discharging process under these conditions occurs at a comparably slow rate of about two and eight hours, respectively (Supplementary Fig. 6B, C).

As previously shown for a Ru-POM dyad and in line with thermodynamics, on-demand hydrogen evolution can be facilitated by increasing the proton concentration[7]. This is especially true for derivatives of the employed rhodium catalysts, as the catalytically active hydrogen evolving hydride species is only formed below a certain pH

value[41]. The pK$_a$ value of the two-fold reduced **3** is 3.8 (see Supplementary Fig. 9), revealing that it is the predominant species at a pH value of 2. Therefore, in all further experiments the solutions were acidified to obtain a final pH value of 2 using HCl prior to the addition of one of the different HECs. Gas chromatography was able to confirm, that at this pH value all of the employed HECs are active and yield hydrogen. Furthermore, it was verified that the addition of HCl in absence of catalyst did not lead to any hydrogen formation (see Table S1).

As shown in Fig. 3, after acidification of the solutions, the addition of different HECs led to an efficient discharging of the photochemically reduced polymer. However, depending on the selected HEC, the discharging kinetics varied significantly. Whereas the Pt-based HECs (**1** and **2**) discharged the polymer almost completely within the first 20 min (see Fig. 3 for **2**), much slower discharging was observed in the case of the rhodium-based catalysts (see also Supplementary Fig. 10). **1** hereby outpaces the colloidal platinum, which is again indicative of the in-situ particle formation described above. These findings are also in line with the redox potentials of the respective catalysts in comparison to the reduced polymer and the resulting driving forces. The offsets in the degree of charging at $t = 0$ are the result of the fast discharging occurring between sample preparation and measurement of the first data point.

When comparing the two rhodium derivatives with each other it is worth noting that the tbbpy ligand leads to an increased electron density at the Rh center, whereas the electron withdrawing ester groups at the dceb ligand facilitate the reduction of the catalyst by the reduced polymer. The respective Rh(III)/Rh(I) reduction potential of −0.9 V vs. Fc$^+$/Fc[36,37] of **3** leads to the superior catalytic properties in this system, compared to the tbbpy derivative with a reduction potential of −1.1 V vs. Fc$^+$/Fc[37,38]. This also explains the fact that **4** is incapable of discharging the polymer or does so only very slowly, depending on the pH value of the solution. Although this tbbpy derivative is able to discharge the polymer under acidic conditions, the kinetics differ significantly from the other HECs. Rather than the exponential decay described for **2** and **3** in Fig. 3, the [(tbbpy)Rh(Cp*)Cl]Cl catalyst showed a slow and linear discharging process. This was revealed by the steady discoloration, also observed by UV/vis spectroscopy (Supplementary Fig. 10D), and correlates with comparatively low catalytic activity (see below).

After the addition of the different HECs to the acidified polymer solutions, hydrogen evolution was determined using gas chromatography. Correlation of gas chromatographic analysis of hydrogen present in the headspace of the different samples and determination of the degree of charging via UV/vis absorption spectroscopy prior to catalyst addition, furthermore enabled the calculation of the

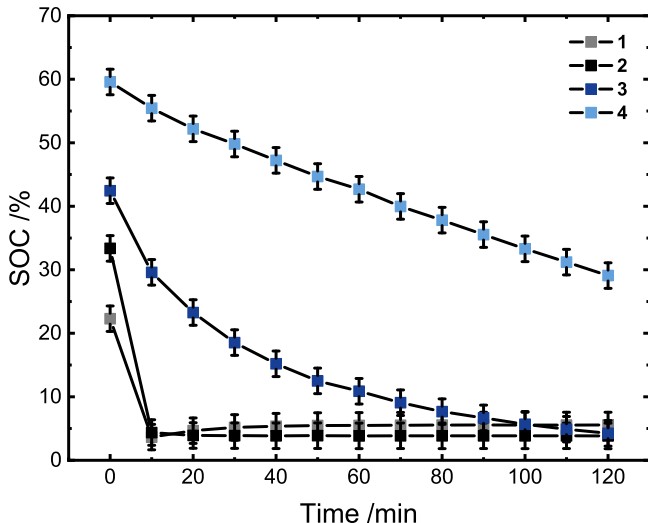

**Fig. 3 | Discharging process of the polymer.** Cross-sections of UV/vis spectroscopic changes of photochemically reduced polymer (450 μM with respect to methyl viologen monomer content, 1 h of irradiation), using [Ru(tbbpy)$_3$]Cl$_2$ (25 μM) in water containing 0.18 M TEA and 0.15 M NaH$_2$PO$_4$ upon addition of 20 mol% (with respect to 450 μM methyl viologen monomer units) of the respective HEC following acidification of the sample to a pH value of 2 using HCl (2 M). SOC refers to state of charging; mean values are displayed and error bars represent standard deviation of at least $n = 3$ independent measurements. Source data are provided as a Source data file.

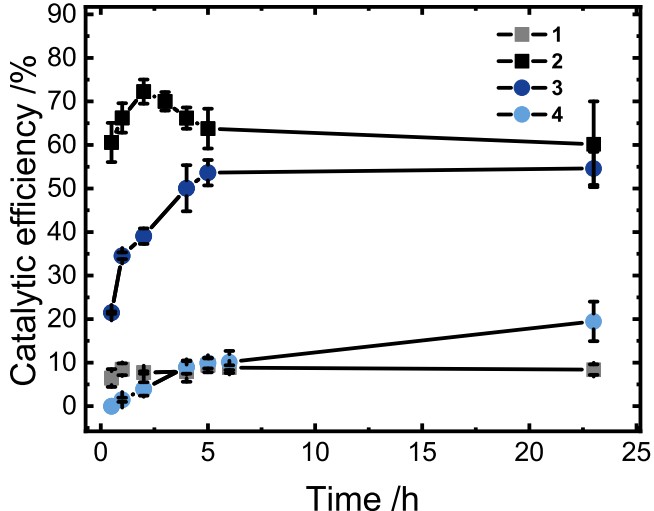

**Fig. 4 | Catalytic performance of HECs.** Catalytic efficiency calculated for each hydrogen evolving complex over time. Mean values are displayed and error bars represent standard deviation of $n = 4$ independent measurements. Source data are provided as a Source data file.

efficiency of the conversion of the polymer-stored electrons into hydrogen gas.

In hand with the faster discharging of the reduced polymer by **2** compared to **3**, the highest conversion efficiency of electrons into hydrogen was obtained for **2** within 2 h, whereas hydrogen evolution using **3** increased over the first 4–5 h. For **2** a maximum conversion efficiency for electrons stored in the redox-active polymer to hydrogen of up to $72 \pm 3\%$ was found, whereas a value of $54 \pm 3\%$ was found for **3**, as shown in Fig. 4 (see SI for a detailed description on the calculation of efficiencies). Possible side reactions leading to the non-perfect conversion of electrons to hydrogen such as reduction or hydrogenation of the polymer were ruled out by UV/vis absorption studies of the

uncharged polymer in the presence of an excess of hydrogen under catalytic conditions with no visible alteration of UV/vis signatures over extended periods of time (see Supplementary Fig. 8).

Notably less efficient hydrogen evolution was found for **1** and **4**, despite the fast discharging of the polymer by the former. For **1** this is likely associated with the initial conversion of Pt(II) to Pt(0) species, most likely accounting for the fast discoloration and the associated consumption of a large number of reducing equivalents (40% SOC would be consumed for all Pt(II) to Pt(0) reductions) stored on the polymer. Contrary, **4** possessing a cathodically shifted Rh(III)/Rh(I) potential compared to **3** evolves hydrogen less efficient than the ester-substituted Rh complex which is easier to reduce by 0.2 V. It provides a continuous discharging course with a maximum conversion of only $10 \pm 3\%$ within the first 6 h of catalysis; **1** reached a similar maximum conversion of $9 \pm 3\%$ over the same period of catalysis. In comparison, experiments employing the best-performing **2** to discharge molecular viologen were performed as well, revealing very poor conversions (see Table S2), which represents only 37% of the turnover, that the polymeric system is able to reach under identical conditions thus highlighting the beneficial properties of the polymer for the presented application.

### Preservation of the charged species

Polyviologens as well as low molar mass viologens have been utilized as active materials in organic flow batteries. In these systems they have proven to be quite stable active materials[30,42]. Consequently, the herein utilized system was also expected to be capable for long-term electron storage. To provide the respective evidence, the polymer was charged, and absorption spectra were measured after a dark period of 24 and 72 h. Promisingly, for the photochemically reduced polymer, no UV/vis spectroscopic changes were observed during the dark period, that could have been assigned to degradation as shown in Fig. 5. Similar stabilities have been reported for molecular MV model compounds of the herein used polymer[26]. Under the established conditions, however, molecular viologen does show vastly inferior storage capabilities, with the molecular viologen losing 40% of the stored charge within the first 18 h of storage (see Supplementary Fig. 12).

Bringing this into a context with other energy storage materials, one can calculate the electric charge that is preserved on a specific amount of polymer (detailed calculation in the SI). Overall this represents a storage of 101 C g$_{Polymer}^{-1}$ for multiple days without loss, compared to similar state of the art systems employing metal organic frameworks (MOF), with storage capacities of 15 C g$_{MOF}^{-1}$[9].

Following up on these results, catalysis measurements of samples that were stored for different periods of time were conducted. In addition to 24 and 67 h storage, catalysis was also initiated after only 2 h of storage. Calculations based on the respective TONs revealed a loss in catalytic efficiency, compared to instantaneous charge retrieval through hydrogen evolution, initiated by **2** directly after photocatalytic charging of the polymer. After storage for 2 h, the efficiency was reduced to $58 \pm 4\%$ and to $56 \pm 9\%$ after 24 h storage as opposed to the $70 \pm 3\%$ reached on average for direct catalytic hydrogen evolution excluding any storage time. Extending the storage period further to 67 h only drops the catalytic efficiency to $48 \pm 4\%$. This means that not only is on-demand hydrogen evolution possible, but also storage of reducing equivalents in the polymer beyond the first two hours entails minimal loss of catalytic activity (0.16% per hour of storage after the initial loss in the first 2 h; Supplementary Fig. 13). Hereby **2** was chosen due to the high turnover frequency (TOF) and high catalytic activity, facilitating a solid readout after a comparably short period of time.

### Regeneration of the system for recycling

As detailed above and due to the polymer´s stability in the range of applied pH conditions (neutral for charging, acidic for discharging) and the known application potential of the polymeric species in redox

flow batteries with multiple charging and discharging cycles[22], also the photochemical storage of electrons in the redox-active polymer might be reversible. Therefore, the herein present system might not be limited to a single catalytic cycle but could also be utilized for subsequent additional ones.

Considering the pH-requirements necessary for discharging of the polymer in presence of the various HECs and also the observations that in presence of **1** irradiation of the solutions can lead to charging of the polymer (see Supplementary Fig. 7), we were wondering if the system as it stands (with PS and HEC both being present in solution) can simply be reused by adjusting the pH value accordingly and without the need to isolate the redox polymer. To follow this path, samples of the two best performing catalysts (**2** and **3**) were prepared analogously to previous measurements but were followed by a neutralization of the acid which was added to induce on-demand hydrogen evolution. Charging without this neutralization step has been proven impossible due to complete protonation of the utilized electron donor TEA (see Supplementary Fig. 11). However, after neutralization with NaOH, it was possible to recharge the polymer photochemically as depicted in Fig. 6.

Direct comparison of the first two charging processes reveals very similar kinetics, with the second process starting at a slightly elevated SOC and reaching a very comparable final charge. Therefore, the charging profile of the second charging step possesses a slightly lower slope and stays within the boundaries set by the preceding charging process. For these results to be as representative as possible for a potential loss in activity, rather than a mismatch in pH, these values were measured under quasi in-operando conditions. With this control over pH, a second cycle was able to almost replicate the initial $71 \pm 2\%$ charge with its $67 \pm 2\%$ SOC.

While charging appears to be stable for at least a second cycle, GC analysis revealed hydrogen amounts that translate to maximum catalytic efficiencies of $55 \pm 4\%$ for **2** and $48 \pm 4\%$ for **3**. This equals to a decrease of 15% for the Pt catalyst, but only a loss of efficiency of 5% for the Rh catalyst. Error margins in Table 1 represent standard deviation of at least $n = 3$ independent measurements.

Experiments investigating a third and fourth reaction cycle were performed as well. In general, the trend indicated by the results of the first two cycles remains, with regards to loss of charging activity and catalytic efficiency (see Fig. 7). Even after three whole cycles, the system can reach an SOC as high as 68%, while catalytic efficiencies decrease steadily and to a higher degree. Catalytic activity of the rhodium catalyst was reduced to 23%, while the efficiency of the platinum catalyst was reduced to 32%, representing a drop in efficiency to less than half their initial activity (see Fig. 7). However, these findings serve as a confirmation that multiple cycles are possible, and they also reveal that the pH-value appears to be not only a switch, but a major factor to tune the system in the future.

To get more insights into potential loss channels, we devised an experiment to simulate the ionic strength of a fourth cycle as starting conditions for both processes. Therefore, the respective amount of sodium chloride is utilized, as it is the source for both ions present in the system. The measurements show that an increase in ionic strength does reduce the systems capabilities to be charged, yielding an explanation for the slight, but steady loss in charging efficiency (see Supplementary Fig. 14). The catalysis on the other hand still reaches efficiencies of $68 \pm 1\%$ for **2**, indicating that this catalyst is not degrading due to ionic strength, but rather due to repeated pH-changes. For **3**, this pH-induced degradation was confirmed by performing resonance Raman measurements of the catalyst exposed to the conditions of the acid base cycle during catalytic charging and

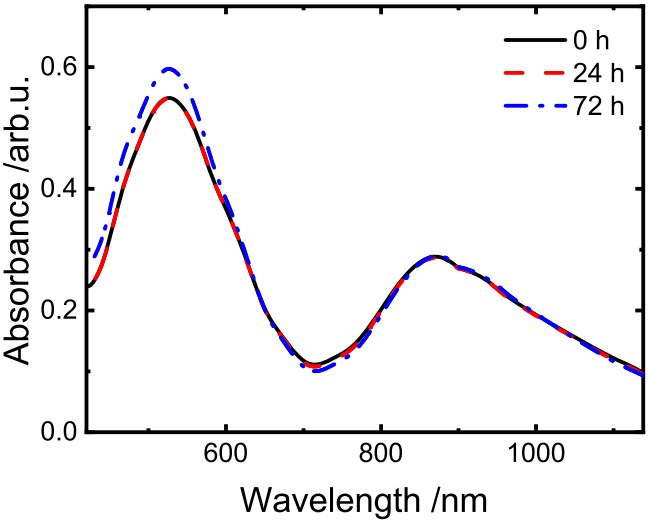

**Fig. 5 | Storage capabilities of the polymer.** UV/vis absorption spectra of the charged polymer (450 μM with respect to methyl viologen monomer content) in an aqueous 0.15 M NaH$_2$PO$_4$ buffer and with 0.18 M TEA directly after the charging process and after 24 and 72 h of storage under inert conditions. Source data are provided as a Source data file.

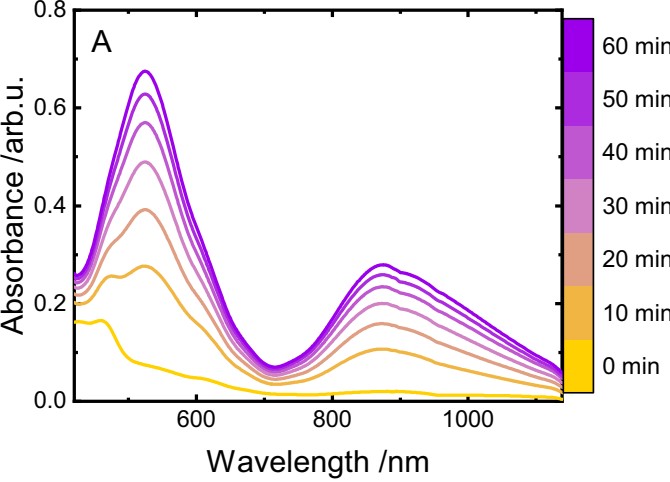
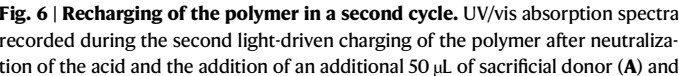
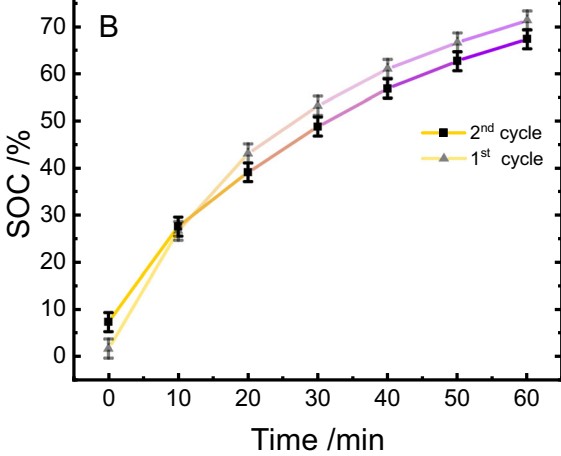

**Fig. 6 | Recharging of the polymer in a second cycle.** UV/vis absorption spectra recorded during the second light-driven charging of the polymer after neutralization of the acid and the addition of an additional 50 μL of sacrificial donor (**A**) and the resulting charging kinetics (**B**). SOC refers to state of charging; mean values are displayed and error bars represent standard deviation of $n = 3$ independent measurements. Source data are provided as a Source data file.

discharging (see Supplementary Fig. 17). It could be shown that upon addition of acid, already at the first discharging step the ester functionality at the bipyridine was cleaved. From cycle 2 onwards, $H_2$ evolution is thus no longer catalyzed by **3** but by a structurally altered bipyridine-based Rh complex.

While this grants an overview of the stability and repeatability of both processes throughout multiple recycling steps, the respective cumulative values should be highlighted. Compiling these numbers for **2**, as in Fig. 8, the major advantage of this system compared to non-regenerable system becomes apparent. Already with four cycles under current conditions, cumulative efficiencies reach the 2.69-fold outcome compared to a theoretical, loss-free single charging process. The catalytic process behaves similarly, with a cumulative catalytic output that reaches 2.06-times the catalytic turnover of a theoretical single-cycle system at complete catalytic conversion. Thereby an ideal single-cycle system is exceeded at least twofold by this multi-cycle system, with the data suggesting not only the polymers stability, but also it not being the bottleneck of the system.

In summary, this represents the first organic polymer-based system, that can be photochemically charged and subsequently drive on-demand hydrogen evolution, while also possessing the regenerative capabilities to enable recycling. Core of this aqueous system is a copolymer consisting of a solubility enhancing as well as a methyl viologen monomer, making it a fully water soluble redox-active soft matter matrix for the described charge storage application. The [Ru(tbbpy)$_3$]Cl$_2$ PS can charge up to 82% of accessible low molecular weight viologen units upon irradiation, thereby exceeding comparable Re-containing MOF-based systems more than six-fold with respect to its gravimetric energy density[9]. The charged state is easily preserved for at least three days and can be consumed in a hydrogen-evolving dark reaction at any given time. Initiated by acidification, the polymer is discharged by a suitable catalyst, with a catalytic conversion efficiency of up to 72%.

Inversely and due to the component´s stability within the range of applied pH conditions, returning to neutral pH value enables repetitive light-driven charging, establishing the systems unique regenerative capabilities. It is shown that a mere pH value switch allows for multiple charging and on-demand catalysis cycles.

## Methods
### Materials and measurements
Unless otherwise stated, all chemical compounds were obtained from commercial sources and used without further purification. Solvents used for synthesis were of technical grade, while measurements were performed in HPLC grade solvents. $^1$H-NMR-analyses employed a Bruker Avance 400 MHz device at ambient temperature.

### Absorption spectroscopy
UV/vis-absorption spectroscopy was performed on a JASCO V-670 UV-vis-NIR Spectrometer or an Avantes AvaSpec-ULS2048CL detector unit coupled with an AVA AvaLight-DH-S-BAL light source, using gas-tight quartz cuvettes ($d$ = 10.0 mm, Starna).

**Table 1 | Overview of averaged efficiencies for charging and catalysis for all investigated catalysts including the results of the second cycle for 2 and 3**

|   | 1st charging | 1st catalysis | 2nd charging | 2nd catalysis |
|---|---|---|---|---|
| **1** | 62% ± 2% | 9% ± 3% | - | - |
| **2** | 79% ± 2% | 72% ± 3% | 69% ± 2% | 55% ± 4% |
| **3** | 78% ± 2% | 56% ± 3% | 78% ± 1% | 48% ± 4% |
| **4** | 62% ± 2% | 19% ± 3% | - | - |

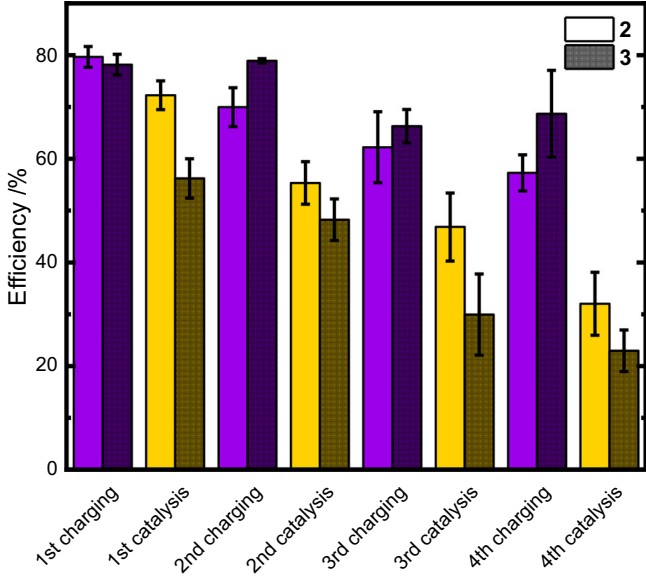

**Fig. 7 | Four-cycle overview for top catalysts.** Chart of efficiencies over four consecutive charging (purple) and catalysis (yellow) cycles employing **2** (Pt-NPs) and **3** ([(dceb)Rh(Cp*)Cl]Cl). Mean values are displayed and error bars represent standard deviation of at least $n$ = 3 independent measurements. Source data are provided as a Source data file.

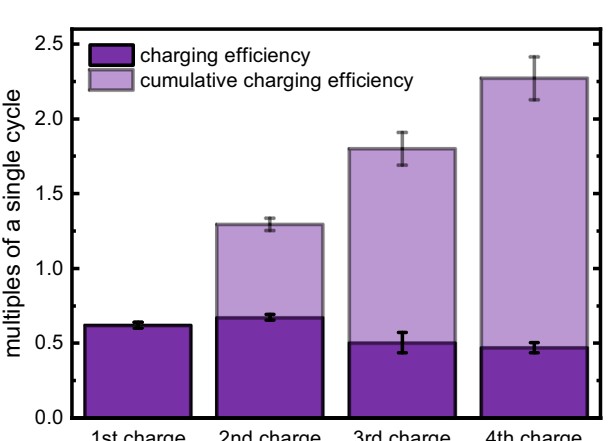
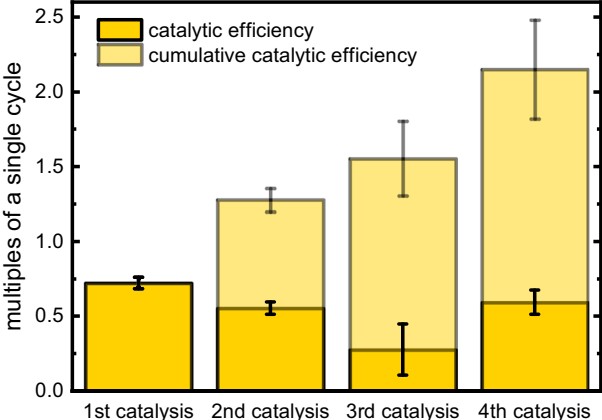

**Fig. 8 | Cumulative charging efficiency and catalytic performance.** Charts of cumulative efficiencies for charging (left) and catalysis (right) process for **2**. Mean values are displayed and error bars represent standard deviation of at least $n$ = 3 independent measurements. Source data are provided as a Source data file.

## Gas chromatography

The amount of generated hydrogen/methane/carbon monoxide was determined by gas chromatography (GC) on a Shimadzu GC-2030 with a barrier ionization discharge (BID-2030) detector and helium as carrier gas (column: Restek SH-Rt-Msieve 5 A, ID: 0.32 mm; film thickness: 30 μm, length: 30 m, oven temp. 80 °C) using 100 μL of the gas phase. The GC was calibrated by injection of different volumes of a test gas mixture containing known percentages of hydrogen/methane/CO.

## Raman and resonance Raman

Raman measurements were performed at a MultiRam Fourier-Transform Raman-Spectrometer (Bruker Corporation, Billerica, Massachusetts, United States of America) with a spectral resolution of 4 cm$^{-1}$ in the range 0 and 4000 cm$^{-1}$. The Raman excitation light at 1064 nm was provided by a Nd:YAG laser (Klastech DeniCAFC-LC-3/40, Dortmund, Germany). The laser power incident upon the sample plane was approximately 250 mW with 100 scans per sample. Background correction was performed using SNIP function (iterations: 200, order: 3, smoothing window: 1). Thereafter, the spectra were confined to the spectral range of 100–4000 cm$^{-1}$ and normalized using vector normalization to minimize variability in intensity of the samples.

In operando Raman spectroscopy was conducted using a NIR – excited fiber-coupled Raman setup (RXNI, Kaiser Optical Systems, Ann Arbor, MI, USA). The sample was excited with a power of 250 mW at the sample through a fiberoptic probe (InPhotonics, Norwood, MA, USA) which is equipped with band and long pass filters for cleanup and Rayleigh-rejection (cutoff long pass: ca. 250 cm$^{-1}$) with a focal spot width of about 50 μm which is coupled to a 785 nm single- frequency laser diode (Xtra II, Topica Photonics, Munich, Germany). This system disperses the scattered Raman light passing through a holographic grating and then detected via a multi-row, thermoelectrically cooled ($T_{OP}$ = −60 °C) CCD (Andor, Belfast, UK) resulting in spectral resolution around 4 cm$^{-1}$. The system was white light calibrated using a white light source prior to the measurements.

Resonance Raman measurements of intermediate states, were carried out using an excitation wavelength of 643 nm of a diode-pumped solid-state laser (CrystaLaser, USA). The laser power at the sample was reduced to approximately 12 mW to minimize photodegradation of the Rh$^I$ chromophore. Raman signals were collected using an IsoPlane 160 spectrometer (Princeton Instruments, USA), with a 30 μm entrance slit and grating with 1200 grooves/mm. The spectrometer is equipped with a thermoelectrically cooled CCD camera (PIXIS eXcelon, Princeton Instruments, USA) featuring a resolution of 1340 × 100 pixels. The H$_2$O spectral band at 1633 cm$^{-1}$ was utilized as a reference for normalizing both intensities and wavenumbers. For spectral analysis, rR spectra were background-corrected, and the solvent spectrum was subtracted.

## Statistics and reproducibility

Error margins featured on the respective experiments represent the standard deviation, unless stated otherwise. These result from either triplicates, or two independent duplicates. The only data excluded in these cases is data acquired during early optimization of the respective experiments, or in clear cases of an experimental error. No statistical method was used to predetermine the sample size. The experiments were not randomized. The experimenters were not blinded to allocation during experiments and outcome assessment.

# Data availability

The authors declare that all data supporting the findings of this study are available within the paper and its supplementary information files. Source data for figures are provided with the paper. Source data are provided with this paper.

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

## Acknowledgements

Financial support by the Deutsche Forschungsgemeinschaft (DFG, German Research Foundation)—Projektnummer 364549901 – TRR 234 [A1 (B. D.-I., S.R.), B2 (U.S.S., S.R.), C1 (B. D.-I.), C2 (J. P.)] is greatly acknowledged.

## Author contributions

M.H. synthesized the molecular compounds, conducted all catalysis-related measurements, and wrote the manuscript. R.K. synthesized the polymer. F.M. assisted with the photocatalysis experiments. L.Z. performed and interpreted resonance Raman measurements, while A.E. conducted and analyzed Raman measurements. P.R., S.Z., and M.D.H. contributed to the design of the research. A.K.M. assisted in research design and manuscript preparation. B.D.-I., M.S., and J.P. designed the (resonance) Raman experiments and supported data interpretation. U.S.S. and S.R. conceived and supervised the research. All authors discussed the results and revised the manuscript.

## Funding

## Competing interests

The authors declare no competing interests.
