## [Transparent Peer Review file · Nature Communications]

A Water-soluble Copolymer for Storage and Electron Conversion in Photocatalytic On-demand Hydrogen Evolution

Corresponding Author: Professor Sven Rau

Version 0:

Reviewer comments:

Reviewer #1

(Remarks to the Author)

In this manuscript, the authors present a reusable system for 'on-demand hydrogen production' with a molecular approach. In this system, charge is stored in a previously studied styrene-based viologen polymer system and its discharge to H₂ is studied with four different types of catalysts—two molecular Rh systems and two Pt systems that appear to be variants of nanoparticle suspensions.

Generally speaking, the work has been carried out to a high technical standard. The results are quite reasonable in my view, and reflect both ingenuity and the difficulty of building a robust and practical on demand H₂ release system. My concerns with this paper stem from these difficulties, in the sense that I have read the paper with a focus on identifying the pitfalls of the system for future improvements. This is certainly only a model system, in that I anticipate there is a low technological readiness level associated with this work. Therefore, my review is focused on helping the authors to add clarity about pitfalls in the performance of their system—building on their existing manuscript that does indeed show that the system can work in a multicycle fashion to some degree.

I anticipate that this paper could have an impact in the scientific community. As the authors include in their citation list, there are a number of groups that have reported or are working actively on 'on-demand' fuel production systems that could alleviate the difficult issue of H₂ storage in renewable energy schemes.

I anticipate that addressing my recommendations could require a round of major revisions to improve a few aspects of the paper and correct misconceptions. These include some aspects of the statistical work which are claimed on technical replicates. I have included specific points below for my concerns, and I emphasize that these are intended to be helpful to the authors in increasing the quality and impact of their work. Not all may be addressed, but I suggest they be at least considered.

I thank the authors for their generally careful work and wish them success in the future.

Specific points

1. At several points in the manuscript the authors claim that their error bars represent standard deviation of at least $n = 2$ independent measurements. Having $n = 2$ with computation of an error bar in this way is not standard practice in the field and I have not previously encountered it in reliable high-quality papers. For all the data points that reflect replicates of $n = 2$, I recommend that an additional replicate be executed so that $n = 3$. From my viewpoint as an inorganic chemist with significant experience in quantitative data analysis, it could attract significant unwanted attention to have n equal to only 2; under such conditions, the estimated standard deviation has little value, in that there is no measure of whether the distribution of values is within what would be expected for the normal distribution of values. Considering the somewhat variable behavior observed later in the paper (see point 16 below), I suggest high caution here. At least, the error bars for points with $n = 2$ can be removed and only the averaged data point for the two individual replicates can be retained.

2. In the introduction, the authors motivate use of a polymer system by referring to 'nonproductive light-induced processes of

fully molecular systems.' What is a process of this type that would have occurred in the present system without the polymer? Or, what is an example of one in general? I ask because this system appears sensitive to unproductive quenching and also unproductive dimerization of viologen. Thus, the question could be asked what the polymer does here. Also, there is no control in this paper for use of viologen and Ru(bpy)₃ without polymerization...so that could be considered for addition too to properly motivate and control for use of the polymer.

3. Please describe what the styrene-viologen polymer was previously used for in reference 22. This should be mentioned in the paper. I suppose it may have been used for electroanalytical work with immobilized redox enzymes...but the authors could usefully share this with readers for context.

4. Is the UV-visible spectrum of the charged viologen polymer consistent with dimerized viologen? This should be known and could be checked, as discussed on p 4.

5. There are multiple broken references that are associated with the text in German "Fehler! Verweisquelle konnte nicht gefunden werden."

6. Why can the polymer only reach 60% charging? Also, how was this level determined? Was elemental analysis used to measure the amount of N (viologen derived) in the solid material? Or, was a spectroscopy assuming similar properties to monomeric models used? This should be specified. Also, does the 60% level indicate that some of the viologen remains undimerized? And finally, does absorption of light by the apparently darkly colored, reduced viologen preclude light absorption by the Ru(bpy)₃ system contribute to the lower than perfect charging? I realize the latter might be challenging to confirm but it strikes me as an important point.

7. How was the 1:1 viologen:styrene ratio confirmed? I assume a measurement of this was done rather than simply relying on a literature result.

8. When a strong chemical reductant is used under air- and moisture-free conditions, can all of the viologen be reduced in the polymer? Answering this question quantitatively is not strictly required for understanding behavior with Ru-based reductants, but I ask because it speaks to the efficacy of this polymer in the tested conditions. What would prevent full reduction, i.e. full efficiency in this system?

9. What are the reduction potentials of the Rh catalysts relative to the viologen and Ru(bpy) on a common potential reference? This should be updated as presently the Rh is quoted vs. Fc^{+/0} and the viologen isn't specified I believe. This is important because it appears that the tbu bpy system cannot be easily reduced by the polymer resulting in very sluggish H₂ evolution. The more easily reduced ester bpy can however be reduced.

10. Does the quotation of 101 C per gram polymer on p. 9 take into account the incomplete charging found in the Ru-bpy work?

11. The colors of the data markers in Fig 3 are too similar to be easily understood by readers.

12. Does the Pt NP system take up H₂ and hydrogenate the polymer? In other words does the Pt take up added H₂ with the oxidized polymer and reduce it or hydrogenate some functionalities? Where would the H₂ generated in the data presented be going since it is apparently lost from the flask upon generation over 24 h? This behavior should be explained. If the system is being hydrogenated, it would suggest limited usefulness with HEC2, which appears later in the recyclability section.

13. How was the polymer stored upon charging but before the H₂ release as described on p. 10. Was it left in the presence of H₂O and more importantly O₂? I suppose O₂ could scavenge reducing equivalents. Was the polymer ever isolated as a solid or was it always kept in solution?

14. Plotting H₂ released as a function of time delay after charging could be useful for readers as a figure in main text or SI. It appears from the data available so far that there is a rapid decrease in performance and then a slower phase as well.

15. Does the pH swing system required for function of this polymer mean that ionic strength was not constant in solution over the multi-cycling experiments? Was any control done to test for an influence of ionic strength on charging and H₂ evolution efficiency, in the testing described on p. 11 or so?

16. The 'small scale approach' and 'intrinsic requirements of the reaction system' are described as reasons for the variable and somewhat unexpectedly variable results across multicycling on p. 12. These reasons/speculations are not clear or appropriate. Are the authors saying that the experiments are so difficult that they cannot be done with reliable analytical quality? Please revise this section. Perhaps more technical replicates are required to establish reliable performance metrics of this system?

17. HEC3 appears non-recyclable as it starts to behave quite poorly for hydrogen release in Fig. over 4 cycles of use. Is this because of degradation of the catalyst?

18. What cycling time was used for each stage of the multi-cycling? I ask because HEC2 is the high performer, but its performance was clearly time dependent in Fig 4. The time utilized for the H₂ release should be noted carefully in the work

on p 14 for the multicycling efficiency test. This is because the time for the H₂ cycle should make a large difference in the metrics achieved for the multi-cycle efficiency, right?

Reviewer #2

(Remarks to the Author)

The authors have reported the first system based on organic polymers that can be charged photochemically and subsequently drive hydrogen evolution on demand. Additionally, this system possesses regenerative capabilities that enable recycling and multiple catalytic cycles. In the presence of triethylamine as a sacrificial donor, a charging efficiency of over 60% can be achieved, which remains stable for multiple days. By subsequent acidification and the addition of different catalysts, the stored electrons can be utilized on demand for proton reduction to hydrogen with an efficiency of up to 72%. Overall, this work is intriguing and the manuscript is well-written. However, there are some important issues that need to be addressed, which can help largely increase the impact of this manuscript.

1. It is good to design catalysts and obtain high activity. It would be even better if meanwhile new catalysts except for HEC1, HEC2, HEC3, and HEC4 could be offered.
2. Although redox-active polymer synthesis has been previously reported in literature, more detailed experimental procedures and characterization results should be provided in this manuscript.
3. Please provide a better explanation regarding the novelty of the water-soluble copolymer system.
4. It would be beneficial to discuss the stability of these catalysts and redox-active units as well as their potential application in other catalytic reactions.
5. The significant variation observed in discharge kinetics when different HEC is selected needs further clarification.
6. In Fig. S1, it appears that there is not much difference between the degree of charging at pH = 7 compared to pH = 8 but it differs greatly from other pH conditions. Please explain why this discrepancy exists.
7. Please provide evidence supporting enhanced solubility of monomers and copolymers as well as proof for formation of a completely water-soluble redox-active soft matter matrix within your manuscript.

Version 1:

Reviewer comments:

Reviewer #1

(Remarks to the Author)

The revised version of this paper has addressed virtually all of my concerns from the prior round of review. Concerns related to the number of replicate experiments have been addressed through both a greater number of replicates as well as improved analytical methods. Other points have been addressed as well in ways that I view are acceptable. Needed additional data to understand the context of the results have been included in the revised version as well. In short, this paper now appears ready for publication and I suggest that it be accepted.

One small question that occurred to me after re-reading the materials was the question of degradation of the rhodium molecular catalyst over multicycling. Was characterization of the Rh component done after one or more cycles of catalysis? If so, the results being added to the SI would be recommended. Maybe the ¹H NMR of the catalyst is available and can be compared before/after run(s)?

Reviewer #2

(Remarks to the Author)

In the revised manuscript, the authors have properly addressed the concerns from previous reviewers. Therefore, I would like to recommend publication of this manuscript at its current version.

Version 2:

Reviewer comments:

Reviewer #1

(Remarks to the Author)

In the revised version of the manuscript, the authors have provided a quite thorough reply to my prior comment requesting that the fate of the molecular catalyst be examined. Through very detailed analysis involving several techniques, the authors have identified a degradation pathway of their catalyst involving hydrolysis of ester groups under low-pH conditions. I emphasize here that the identification of this degradation pathway in no way tampers my enthusiasm for this report—rather,

the very thorough and chemically sensible conclusions drawn by the authors add significantly to the quality of the science presented here. It is a lamentable draw-back of many chemical studies of molecular catalysts for energy conversion and storage that degradation pathways and longevity studies are not studied in more detail--this study now overcomes this common problem. I have reviewed the new data from Raman studies provided by the authors and agree with their interpretations.

In terms of the path to wrap this up, I have only a single final suggestion for their consideration. On page 2 of their response document, they refer to a $d8$ to π^* MLCT for the Rh(I) species. The noted spectral features between 540 and 840 nm, from what I know of species of this type, given them an appealing dark coloration (in derivatives that I have studied, purple). However, rather than just being a $d8$ to π^* transition, this coloration can be also thought of arising from mixed $[Rh(I) d8 + bpy \pi^*]$ to $[\pi^*]$ transitions, owing to delocalization of charge at the Rh(I) state into the first LUMO of the bpy system. The evidence for this comes often from vibronic features that, granted, are not obvious in the new spectra in Fig S16 and shown on p. 3 of the response document. So, considering all this, the authors may wish to adjust their interpretation/description of the origins of the new beautiful absorptions slightly. This is optional however and I emphasize that this does not need to be reviewed by me again.

This is a lovely contribution to the literature of Rh catalysis, energy conversion, and the opportunities and challenges of building systems for energy storage in chemical bonds. I congratulate the authors on their exceptionally beautiful and thorough work. And, I look forward to seeing it in print for all to read very soon.

Reviewer 1:

In this manuscript, the authors present a reusable system for 'on-demand hydrogen production' with a molecular approach. In this system, charge is stored in a previously studied styrene-based viologen polymer system and its discharge to H₂ is studied with four different types of catalysts—two molecular Rh systems and two Pt systems that appear to be variants of nanoparticle suspensions.

Response: Within this work two different molecular Rh catalysts have been used as well as two different Pt systems. In one of the Pt systems, the nanoparticles have been used as commercially received whereas in the other version a Pt salt was added that was converted into Pt nanoparticles *in situ* using the electrons that were stored at the polymer.

Generally speaking, the work has been carried out to a high technical standard. The results are quite reasonable in my view, and reflect both ingenuity and the difficulty of building a robust and practical on demand H₂ release system. My concerns with this paper stem from these difficulties, in the sense that I have read the paper with a focus on identifying the pitfalls of the system for future improvements. This is certainly only a model system, in that I anticipate there is a low technological readiness level associated with this work. Therefore, my review is focused on helping the authors to add clarity about pitfalls in the performance of their system—building on their existing manuscript that does indeed show that the system can work in a multicycle fashion to some degree.

Response: We addressed some of the challenging aspects and hope that this version of the manuscript will find a positive response.

I anticipate that this paper could have an impact in the scientific community. As the authors include in their citation list, there are a number of groups that have reported or are working actively on 'on-demand' fuel production systems that could alleviate the difficult issue of H₂ storage in renewable energy schemes.

I anticipate that addressing my recommendations could require a round of major revisions to improve a few aspects of the paper and correct misconceptions. These include some aspects of the statistical work which are claimed on technical replicates. I have included specific points below for my concerns, and I emphasize that these are intended to be helpful to the authors in increasing the quality and impact of their work. Not all may be addressed, but I suggest they be at least considered.

I thank the authors for their generally careful work and wish them success in the future.

Response: In the following, we hope that we can answer any of the questions appropriately. We devised additional experiments, repeated existing ones and altered the manuscript as well as the Supporting Information file at the respective positions. We hope that this could clarify the so far unclear aspects of the manuscript.

Specific points

1. At several points in the manuscript the authors claim that their error bars represent standard deviation of at least $n = 2$ independent measurements. Having $n = 2$ with computation of an error bar in this way is not standard practice in the field and I have not previously encountered it in reliable high-quality papers. For all the data points that reflect replicates of $n = 2$, I recommend that an additional replicate be executed so that $n = 3$. From my viewpoint as an inorganic chemist with significant experience in quantitative data analysis, it could attract significant unwanted attention to have n equal to only 2; under such conditions, the estimated standard deviation has little value, in that there is no measure of whether the distribution of values is within what would be expected for the normal distribution of values. Considering the somewhat variable behavior observed later in the paper (see point 16 below), I suggest high caution here. At least, the error bars for points with $n = 2$ can be removed and only the averaged data point for the two individual replicates can be retained.

Response: The issues with reproducibility at that stage has since been resolved and we are now able to present our revised experiments. With *in-operando* measurements of the pH-value, we are now able to control the cycling procedure in a much more precise fashion, which should also grant more clarity to the systems true capabilities and limitations.

This new method is described in greater detail in the Supporting Information, Chapter 1 and repetitions with said procedure did not only help us meet the scientific standard of triplicates for remaining measurements, but also achieve better reproducibility, seen in Fig. 6 in the manuscript.

On page 4 of the Supporting Information, the optimized protocol is provided:

“The initial dummy system, confirming calculations, refers to an experiment, employing the buffer solution and the addition of respective amounts of acid and base. Depending on the measurement, either the GC-vial (for hydrogen detection) or the inert cuvette (for UV/vis absorption spectroscopy) were sealed immediately afterwards, ready to be removed from the glovebox. Measurements without acidification were also performed (Fig. S7) by skipping the addition of the HCl, maintaining the remaining procedure. In a revised procedure, the dummy system relied on actual samples, rather than an upscaled titration with the buffer solution. To measure the pH-value of the actual sample a needle-pH-meter was employed, capable of piercing the septum of an inertly prepared and gastight sample vial. This method is as close as possible to the experimental conditions, which greatly enhances the precision and control over the pH-value, especially when moving to higher cycle numbers in the multicycle experiments.”

The previous issues on large error bars associated with the individual measurements stem from the way, in which the pH-values of the samples were directed. The amounts of acid and base were calculated and titration experiments with the buffer solution were conducted using a pH-meter. To closer match the experimental conditions, we are now using a needle-pH-meter, which is capable to determine the pH value of the actual samples prepared under experimental conditions. This was especially helpful for the third and fourth cycle, which we have been lacking triplicates beforehand, due to increasing discrepancies to the initial titrations.

2. In the introduction, the authors motivate use of a polymer system by referring to ‘nonproductive light-induced processes of fully molecular systems.’ What is a process of this type that would have occurred in the present system without the polymer? Or, what is an example of one in general? I

ask because this system appears sensitive to unproductive quenching and also unproductive dimerization of viologen. Thus, the question could be asked what the polymer does here. Also, there is no control in this paper for use of viologen and Ru(bpy)₃ without polymerization...so that could be considered for addition too to properly motivate and control for use of the polymer.

Response: With nonproductive light-induced processes of fully molecular systems we referred to already investigated examples in which a dye is covalently linked to an acceptor unit and, thus, in close spatial proximity to it (see Kahlfuss, C. *et al. Comptes Rendus. Chimie* **2014**, *17*, 505–511). It has been found that connecting those units is typically beneficial in terms of forward electron transfer from the dye towards the acceptor side (a catalyst for a reductive process) but this strategy also offers pitfalls. Once the catalyst is reduced one- or two-fold a new excitation of the dye can oxidize the acceptor side which negatively impacts its activity. In order to clarify this aspect, we have rewritten the corresponding sentence in the introduction as followed:

“In addition to time-delayed fuel generation, the interposition of polymeric electron storage materials offers the possibility to circumvent nonproductive light-induced processes of fully molecular systems, such as light-induced intramolecular charge redistributions during and after the formation of reductively activated catalyst sites that negatively influence the system’s activity.”

Also, it should be clarified that the dimerization of the polymer-bound methyl viologens is not considered to represent an unproductive (*i.e.* energy wasting) but rather fully reversible process only resulting from the high local concentration of mono-reduced methyl viologen units arranged at the polymer backbones. Based on UV-vis spectroscopic studies performed in this work the well-known dimerization process does not prevent full polymer re-oxidation upon catalyst and/or acid addition.

Also, resting the comparison on literature might in fact have been insufficient, which is why we implemented experiments with molecular viologen, following your advice. The experiments (carried out under identical conditions) were able to show the superior properties of the copolymer under the conditions utilized herein, in particular when it comes to charge storage as well as charge transfer to a catalyst. The respective data is presented in the SI by **Fig. S12** showing a comparably rapid decrease of charge stored on molecular viologen, as well as in **Table S2**, revealing a catalytic turnover of about a third compared to the polymer.

3. Please describe what the styrene-viologen polymer was previously used for in reference 22. This should be mentioned in the paper. I suppose it may have been used for electroanalytical work with immobilized redox enzymes...but the authors could usefully share this with readers for context.

Response: Its first application was in a redox-flow battery, which is now mentioned in the manuscript as first sentence in Chapter 2.1:

“The redox-active polymer, which was previously used in polymer redox-flow batteries, was synthesized according to literature procedures.”

4. Is the UV-visible spectrum of the charged viologen polymer consistent with dimerized viologen? This should be known and could be checked, as discussed on p 4.

Response: Yes, the described bands are indicative of dimerized viologen, as the polymer possesses sufficiently high local concentrations to form them. We clarified that in the manuscript in Chapter 2.1. and added another reference, to extend on this behavior:

“These bands are indicative of the formation of reduced viologen units as has previously been shown for structurally similar viologen-containing polymers and suitable molecular reference compounds in literature.²⁴⁻²⁶ The broad and rather unstructured UV-vis profile deviates from the finding for molecular methyl viologen (MV²⁺), which exhibits a more fine-structured absorption profile under otherwise identical conditions.^{24,25}”

5. There are multiple broken references that are associated with the text in German “Fehler! Verweisquelle konnte nicht gefunden werden.”

Response: It is a referencing problem and has now been fixed accordingly.

6. Why can the polymer only reach 60% charging? Also, how was this level determined? Was elemental analysis used to measure the amount of N (viologen derived) in the solid material? Or, was a spectroscopy assuming similar properties to monomeric models used? This should be specified. Also, does the 60% level indicate that some of the viologen remains un-dimerized? And finally, does absorption of light by the apparently darkly colored, reduced viologen preclude light absorption by the Ru(bpy)₃ system contribute to the lower than perfect charging? I realize the latter might be challenging to confirm but it strikes me as an important point.

Response: The improved experiments were actually able to reach charging above 80% (as shown in Fig. 1) and potential loss channels are also addressed in the manuscript in the end of chapter 2.4:

“To get more insights into potential loss channels, we devised an experiment to simulate the ionic strength of a fourth cycle as starting conditions for both processes. Therefore, the respective amount of sodium chloride is utilized, as it is the source for both ions present in the system. The measurements show that an increase in ionic strength does reduce the systems capabilities to be charged, yielding an explanation for the slight, but steady loss in charging efficiency.”

Also, we did improve the explanation of how the state of charging was determined in the manuscript (second to the last paragraph of 2.1):

“In order to evaluate the efficiency of this charging process, the absorption of the fully reduced polymer was recorded using chemical reductants. Utilizing this data enables a reference system, allowing a quantitative comparison of the photochemically reduced samples to the chemically reduced standard. The utilized reducing agent Na₂S₂O₄ is capable of fully reducing the redox-active viologen units at various polymer concentrations²⁸ (see SI for a detailed description of the experimental procedure).”

As well as the detailed explanation in the first chapter of the SI:

“With 60 equivalents of dithionite per MV²⁺ unit, measurements of polymer solutions with concentrations ranging from 20 to 120 μM (in 20 μM steps) were performed. The respective absorption maxima enabled a linear fit, which allows the determination of the absorbances of a fully reduced polymer at any concentration. Inversely, the comparison of the absorbance of any samples with a known concentration to this reference enables the determination of the state of charging of the sample at any given time.”

To address the co-absorption of the arising polymer band with the photosensitizer as the potential bottleneck for incomplete charging compared the decreasing charging increments per time with the decreasing percentage of the Ru-MLCT at the irradiation wavelength. This comparison reveals two different diminishing trajectories, which prove that increasing optical density may be part of the reason, but is not the sole reason the polymer can not be fully charged under our conditions, as it is now written at the end of chapter 2.1:

*“Addressing the incomplete charging, it can be noted that the arising absorption of the charged polymer does overlap with the absorption of the photosensitizer. A closer look at the charging trajectory, compared to the percentage the ruthenium absorption constitutes at the irradiation wavelength reveals, that co-absorption of photosensitizer and polymer is not responsible for the incomplete trend of the SOC (see **Fig. S4**). While the co-absorption certainly can be a factor in explaining the SOC limit, data suggests that the bottleneck is more likely to be the result of limited energy transfer.”*

7. How was the 1:1 viologen:styrene ratio confirmed? I assume a measurement of this was done rather than simply relying on a literature result.

Response: It is assessed via NMR spectroscopy in previous studies, as the manuscript will now state in the first paragraph of 2.1:

“The ratio of those two moieties in the final copolymer was 1:1, which was confirmed by NMR spectroscopy in a previous study.²²”

8. When a strong chemical reductant is used under air- and moisture-free conditions, can all of the viologen be reduced in the polymer? Answering this question quantitatively is not strictly required for understanding behavior with Ru-based reductants, but I ask because it speaks to the efficacy of this polymer in the tested conditions. What would prevent full reduction, i.e. full efficiency in this system?

Response: As described above (Question 6) in the clarification of the determination process, an excess of reducing agent was employed to provide a reference system. The state of charging is given in comparison to this chemically fully reduced reference system. Further excess of chemical reducing agent or utilization of even stronger agents can in fact lead to discoloration caused by the formation of MV^0 species proving that before formation of MV^0 all MV units had to be reduced once (MV^0 units in the presence of MV^{2+} would be unstable, due to comproportionation to MV^{1+}). Using more reducing agent in the photochemical experiments (however, thereby increasing the pH-value of the solution) will cause the described irreversible degradation of the polymer due to high pH-value.

9. What are the reduction potentials of the Rh catalysts relative to the viologen and Ru(bpy) on a common potential reference? This should be updated as presently the Rh is quoted vs. $Fc^{+/0}$ and the viologen isn't specified I believe. This is important because it appears that the tbu bpy system cannot be easily reduced by the polymer resulting in very sluggish H_2 evolution. The more easily reduced ester bpy can however be reduced.

Response: The text should now present the redox potential, as well as a more detailed explanation for this in the first paragraph of 2.2:

“The molecular HEC differ in the bipyridine type ligand which determines the redox potential of the rhodium center (HEC3 = -0.9 V vs. $Fc^+/Fc^{33,34}$, HEC4 = -1.1 V vs. $Fc^+/Fc^{34,35}$), thus, allowing for different driving forces of electron transfer between the reduced polymer and the putative catalytic center.”

10. Does the quotation of 101 C per gram polymer on p. 9 take into account the incomplete charging found in the Ru-bpy work?

Response: It does; the exact calculation has been implemented in the SI and can be found in chapter 3.2.

11. The colors of the data markers in Fig 3 are too similar to be easily understood by readers.

Response: Color and markers have been adjusted.

12. Does the Pt NP system take up H_2 and hydrogenate the polymer? In other words does the Pt take up added H_2 with the oxidized polymer and reduce it or hydrogenate some functionalities? Where would the H_2 generated in the data presented be going since it is apparently lost from the flask upon generation over 24 h? This behavior should be explained. If the system is being hydrogenated, it would suggest limited usefulness with HEC2, which appears later in the recyclability section.

Response: Studies do not suggest hydrogenation or recharging of the polymer in the presence of hydrogen and Pt(0) catalyst. With the multi-cycling experiments at hand, it can be further proven that the polymer was not hydrogenated, as it is possible to charge the polymer multiple times to a high degree. Similarly, a reduction of the polymer by the catalyst could not be observed in the discharging process.

Furthermore, an experiment with uncharged polymer and Pt(0) catalyst in the presence of about 50 to 100 times the concentration of hydrogen present during usual catalysis was performed. The absorption over time shows no significant change, indicating charging or hydrogenation of the polymer, which is now stated at the end of chapter 2.2:

“Possible side reactions, reducing or hydrogenating of the polymer were ruled out by absorption studies in the presence of excess hydrogen under catalytic conditions (see Fig. S8).”

13. How was the polymer stored upon charging but before the H_2 release as described on p. 10. Was it left in the presence of H_2O and more importantly O_2 ? I suppose O_2 could scavenge reducing equivalents. Was the polymer ever isolated as a solid or was it always kept in solution?

Response: Water is the only solvent that is employed and the polymer is highly oxygen sensitive, so the samples are handled appropriately in sealed vials when extracting them from the glovebox. The detailed and revised description of the sample preparation is given in the SI in Chapter 1. Furthermore, spectra of a charged sample before and after aerating it has been added to the SI as Fig. S3, emphasizing the immediate re-oxidation of charged polymer by the oxygen, which is now mentioned in the manuscript in the end of Chapter 2.1:

“This charge is lost instantly upon aerating the sample, highlighting the necessity of the inert handling of samples (see Fig. S3).”

We did try to extract the polymer from the solution in earlier stages of the research with little success, due to the small sample size.

14. Plotting H₂ released as a function of time delay after charging could be useful for readers as a figure in main text or SI. It appears from the data available so far that there is a rapid decrease in performance and then a slower phase as well.

Response: We included a graph **Fig. S13** in the SI, which shows the described catalytic loss, depending on the storage time. It also contains a fit to quantify the slower loss period.

15. Does the pH swing system required for function of this polymer mean that ionic strength was not constant in solution over the multi-cycling experiments? Was any control done to test for an influence of ionic strength on charging and H₂ evolution efficiency, in the testing described on p. 11 or so?

Response: The pH-swing resulted in an increasing ionic strength with every cycling process. We therefore devised an experiment simulating the increased ionic strength of a fourth cycle directly (so in a first cycle). The manuscript now features a description of this experiment, as well as an evaluation of these results at the end of Chapter 2.4. The experiment revealed that charging is diminished with increasing ionic strength, while catalysis is not, meaning decreases in catalytic efficiency are the result of degradation due to pH-changes, rather than the ionic strength itself.

*“To get more insights into potential loss channels, we devised an experiment to simulate the ionic strength of a fourth cycle as starting conditions for both processes. Therefore, the respective amount of sodium chloride is utilized, as it is the source for both ions present in the system. The measurements show that an increase in ionic strength does reduce the systems capabilities to be charged, yielding an explanation for the slight, but steady loss in charging efficiency. The catalysis on the other hand still reaches efficiencies of $68 \pm 1\%$ for **HEC2**, indicating that the catalyst is not degrading due to ionic strength, but rather due to repeated pH-changes.”*

16. The ‘small scale approach’ and ‘intrinsic requirements of the reaction system’ are described as reasons for the variable and somewhat unexpectedly variable results across multicycling on p. 12. These reasons/speculations are not clear or appropriate. Are the authors saying that the experiments are so difficult that they cannot be done with reliable analytical quality? Please revise this section. Perhaps more technical replicates are required to establish reliable performance metrics of this system?

Response: With the improved procedure (mentioned above), we were able to clear these issues, enhancing reproducibility. With the pH-value being a key factor in controlling the experimental conditions, the usage of a needle pH-meter and therefore the ability to investigate the pH-value *in-operando* did greatly improve the reproducibility. Especially the multi-cycling with several acidification and neutralization steps posed a major difficulty in the beginning, where the pH-value was determined *via* titrations in a dummy system (for reference, we are employing samples in the μg range, which are very pH-responsive resulting in difficulties for controlling the exact switch beyond a second cycle, under inert conditions and prior to the *in-operando* pH-measurements with a needle-pH-meter).

17. HEC3 appears non-recyclable as it starts to behave quite poorly for hydrogen release in Fig. over 4 cycles of use. Is this because of degradation of the catalyst?

Response: Results so far indicate that degradation due to the constant changes in pH-value is in fact an issue. Again, the updated data shows improved reproducibility, summarized in Fig. 6, which shows comparable recycling capabilities to the Pt catalyst, albeit with slightly inferior catalytic efficiency.

18. What cycling time was used for each stage of the multi-cycling? I ask because HEC2 is the high performer, but its performance was clearly time dependent in Fig 4. The time utilized for the H₂ release should be noted carefully in the work on p 14 for the multicycling efficiency test. This is because the time for the H₂ cycle should make a large difference in the metrics achieved for the multi-cycle efficiency, right?

Response: The description of the experimental procedure given in the first chapter of the SI has been changed to clarify how multi-cycling is performed.

“These measurements were carried out over a time span of up to six hours in fixed, reasonable intervals to ensure comparability among various experiments. Based on these results, catalytic analysis of a cycle beyond the first one, measurements were performed after two hours and after three hours.”

Reviewer 2:

The authors have reported the first system based on organic polymers that can be charged photochemically and subsequently drive hydrogen evolution on demand. Additionally, this system possesses regenerative capabilities that enable recycling and multiple catalytic cycles. In the presence of triethylamine as a sacrificial donor, a charging efficiency of over 60% can be achieved, which remains stable for multiple days. By subsequent acidification and the addition of different catalysts, the stored electrons can be utilized on demand for proton reduction to hydrogen with an efficiency of up to 72%. Overall, this work is intriguing and the manuscript is well-written. However, there are some important issues that need to be addressed, which can help largely increase the impact of this manuscript.

1. It is good to design catalysts and obtain high activity. It would be even better if meanwhile new catalysts except for HEC1, HEC2, HEC3, and HEC4 could be offered.

Response: With the data at hand (initial and revised submission) we completely agree and plan on establishing new and more robust catalysts. Meanwhile, this manuscript will focus on establishing the polymer in this type of chemistry and, thus, focusing on the utilization of established catalysts before taking this next step.

2. Although redox-active polymer synthesis has been previously reported in literature, more detailed experimental procedures and characterization results should be provided in this manuscript.

Response: The synthesis of the polymer is now shortly described in detail in the first chapter of the Supporting Information and a summary is given in Chapter 2.1 in the manuscript:

“In short, the polymer was obtained by a radical copolymerization of a styrene derivative featuring an ammonium group in order to ensure water-solubility as well as a styrene bearing the redox-active unit – the viologen. The latter monomer was employed enabling the electron storage functionality. The polymerization was performed in water. The ratio of those two moieties in the final copolymer was 1:1, which was confirmed by NMR spectroscopy, similar to a previous study.²²”

3. Please provide a better explanation regarding the novelty of the water-soluble copolymer system.

Response: We updated respective text passages to clarify the origin of the polymer and its role so far. In previous studies, the polymer was employed in redox-flow battery applications, as it is now stated in the manuscript in the first paragraph of 2.1. So, while the polymer itself is not new at this point, its application in this on-demand H₂-delivering system is.

“The redox-active polymer, which was previously used in redox-flow batteries, was synthesized according to literature procedures.²² In short, the polymer was obtained by a copolymerization of a styrene derivative featuring an ammonium group in order to ensure water-solubility. As redox-active unit, a viologen-containing monomer was employed enabling the electron storage functionality.”

4. It would be beneficial to discuss the stability of these catalysts and redox-active units as well as their potential application in other catalytic reactions.

Response: New experiments regarding potential loss channels have been added and a revised repertoire of catalysts will be subject of future research and collaborations. An experiment investigating the influence of the increasing ionic strength revealed that the catalysts are not impacted by ionic strength alone, added at the end of Chapter 2.4:

*“To get more insights into potential loss channels, we devised an experiment to simulate the ionic strength of a fourth cycle as starting conditions for both processes. Therefore, the respective amount of sodium chloride is utilized, as it is the source for both ions present in the system. The measurements show that an increase in ionic strength does reduce the systems capabilities to be charged, yielding an explanation for the slight, but steady loss in charging efficiency. The catalysis on the other hand still reaches efficiencies of $68 \pm 1\%$ for **HEC2**, indicating that the catalyst is not degrading due to ionic strength, but rather due to repeated pH changes. For further insights structural analysis of the polymer as well as the catalysts will be subject of future research.”*

Together with the stability of the charging process throughout all four cycles (seen in Fig. 6), it can also be concluded that the polymer, while impacted by an increasing ionic strength is stable within the given pH-range.

5. The significant variation observed in discharge kinetics when different HEC is selected needs further clarification.

Response: Respective paragraphs have been updated and the necessary information has been added for clarification. This includes the addition of previously missing CV data in the first paragraph of 2.2:

*“The molecular HEC differ in the bipyridine type ligand which determines the redox potential of the rhodium center (**HEC3** = -0.9 V vs. $Fc^+/Fc^{34,35}$, **HEC4** = -1.1 V vs. $Fc^+/Fc^{35,36}$), thus, allowing for different driving forces of electron transfer between the reduced polymer and the putative catalytic center.”*

As well as a more detailed description for Fig. 2:

*“As shown in **Fig. 2**, after acidification of the solutions, the addition of different HECs led to an efficient discharging of the photochemically reduced polymer. However, depending on the selected HEC, the discharging kinetics varied significantly. Whereas the Pt-based HECs (**HEC1** and **HEC2**) discharged the polymer almost completely within the first 20 min (see **Fig. 2** Error! Reference source not found. for **HEC2**), much slower discharging was observed in the case of the rhodium-based catalysts (see also **Fig. S7**). **HEC1** hereby outpaces the colloidal platinum, which is again indicative of the in-situ particle formation described above. These findings are also in line with the redox potentials of the respective catalysts in comparison to the reduced polymer and the resulting driving forces. The offsets in the degree of charging at $t = 0$ are the result of the fast discharging occurring between sample preparation and measurement of the first data point.”*

6. In Fig. S1, it appears that there is not much difference between the degree of charging at pH = 7 compared to pH = 8 but it differs greatly from other pH conditions. Please explain why this discrepancy exists.

Response: The electrochemistry of viologens is stable at neutral pH-values. Higher pH-values lead to a degradation of the viologens *via* dealkylation as it now states more precisely in Chapter 2.1 of the manuscript:

*“The system was buffered with NaH_2PO_4 to guarantee structural integrity of the polymer, as it is able to undergo irreversible dealkylation processes at higher pH-values (see **Fig. S1**).^{23,24}”*

The hydroxyl ions will form methanol and the monoalkylated bipyridine is formed. Therefore, the use is limited to close to neutral conditions. Noteworthy, this degradation mechanism is also important for the use of viologens in redox flow batteries (E. S. Beh, D. De Porcellinis, R. L. Gracia, K. T. Xia, R. G. Gordon, M. J. Aziz, *ACS Energy Lett.* **2017**, 2, 639-644).

7. Please provide evidence supporting enhanced solubility of monomers and copolymers as well as proof for formation of a completely water-soluble redox-active soft matter matrix within your manuscript.

Response: As mentioned above, the viologen copolymers has been previously used as active material in polymer redox flow batteries (see our previous work: T. Janoschka, S. Morgenstern, H. Hiller, C. Friebe, K. Wolkersdörfer, B. Häupler, M. D. Hager, U. S. Schubert, *Polym. Chem.* **2015**, 6, 7801-7811 & T. Janoschka, N. Martin, U. Martin, C. Friebe, S. Morgenstern, H. Hiller, M.

D. Hager, U. S. Schubert, *Nature* **2015**, 527, 78-81). Hereby, aqueous electrolytes have been utilized. The copolymer was well soluble in water. For comparison, the viologen homopolymer without the solubility promoting comonomer is still good water soluble, even in the reduced state (radical cation instead of the dication). The observations in the lab are also substantiated by DLS measurements (see *Nature* **2015**, 527, 78-81), in which only small sizes (ca. 2 nm) have been observed. Due to the very small sizes, no larger aggregates like micelles and particles are present. A remark on this previous study has been implemented in the manuscript in Chapter 2.1:

“The redox-active polymer, which was previously used in aqueous polymer redox-flow batteries, was synthesized according to literature procedures.²²”

Reviewer 1:

The revised version of this paper has addressed virtually all of my concerns from the prior round of review. Concerns related to the number of replicate experiments have been addressed through both a greater number of replicates as well as improved analytical methods. Other points have been addressed as well in ways that I view are acceptable. Needed additional data to understand the context of the results have been included in the revised version as well. In short, this paper now appears ready for publication and I suggest that it be accepted.

One small question that occurred to me after re-reading the materials was the question of degradation of the rhodium molecular catalyst over multicycling. Was characterization of the Rh component done after one or more cycles of catalysis? If so, the results being added to the SI would be recommended. Maybe the ^1H NMR of the catalyst is available and can be compared before/after run(s)?

Response: We thank the reviewer for this highly interesting and relevant question. We performed extensive experimental investigations to answer the raised question. ^1H NMR, Raman- and resonance Raman-spectroscopy were used.

The attempts to perform in situ ^1H NMR investigations to identify the fate of the rhodium catalyst were not successful due to the dynamics of the ligand environment at the Rh center in presence of several potential ligands binding to Rh instead of Cl^- . This was the case in solutions containing buffer and sacrificial donor with and without addition of acid that should mimic the conditions of the acid – base swing. As already under these conditions a non-interpretable ^1H NMR spectrum was obtained, further acid/base swings were not examined.

We then turned to Raman spectroscopy as it would be employable in aqueous media, aiming for the carboxylic ester moiety of the bipyridine ligand of the rhodium catalyst. This moiety can clearly be identified by its Raman band at 1726 cm^{-1} which could be measured. However, in a catalytic mixture and upon photocatalytic charging of the polymer new signals corresponding to the reduced polymer were dominating the spectra in the corresponding wavenumber section, thus preventing the identification of the carboxylic ester moiety within the rhodium catalyst and consequently making it impossible to analyze the fate of the Rh catalyst with this method. The SI now includes this study (see Figure S15 and pages S6-S7):

*“In this study the different components were measured, both each one individually and in an upscaled catalytic mixture, as displayed in 0. This reveals, that beyond the bipyridine vibrations, that are shared with the photosensitizer, the C=O double bond vibration of the ester can be observed very well in the individual measurement. In a catalytic mixture, the dominating vibrations are associated with the sacrificial donor and the polymer, due to the differences in concentration (1:5 for **HEC3**:polymer, 1:2000 for **HEC3**:TEA) and the nature of the vibronic signals. Due to the relatively low concentration of the catalyst in the catalytic mixture the identification of the carboxylic ester moiety within the rhodium catalyst was not possible. Nevertheless, in operando nonresonant FT-Raman studies were helpful to enable a reliable band assignment of the resonance Raman modes of the catalyst.”*

Fig. S15 Experimental Raman spectra of **HEC3**, $[Ru(tbbpy)_3]Cl_2$, MV-polymer, NaH_2PO_4 and triethylamine individually in an upscaled catalytic mixture before and after irradiation.

We then decided to utilize resonance Raman (rR) spectroscopy which can be employed for similar rhodium complexes as shown in literature.¹ Chemical reduction of the rhodium catalyst with formate to the Rh(I) state shifts the absorption maximum of the MLCT ($Rh(d^8) \rightarrow bpy-\pi^*$) to the red with a very broad and strong absorption between 540 and 840 nm (see **Fig. S16**). This can be considered a chemical activation step of the Rh complexes for selectively investigating the Rh complexes as well as structural changes to them by rR spectroscopy using excitation in the red part of the spectrum, *i.e.*, with a 643 nm laser. The rR data revealed, that exposing the Rh-catalyst to one or more pH cycles does cause the ester group to be lost, while an intact chromophore is still maintained throughout the entire process as bipyridine vibrations are observed also after the first acid/base swing. Our interpretation is therefore, that the ester moiety is cleaved during the first change of redox state of the rhodium and additionally the alteration of the pH from slightly basic to acidic and back. The respective results and interpretation are now included in the SI (see Figure S17 and pages S7-S8):

Fig. S 16 UV/vis spectroscopic changes of photochemically reduced **HEC3** (150 μM) over 30 min of irradiation, using $[\text{Ru}(\text{tbbpy})_3]\text{Cl}_2$ (42 μM) in water containing 0.09 M TEA and 0.075 M NaH_2PO_4 .

*“Upon reducing the Rh(III) catalyst (**HEC3**) using sodium formate, the formed Rh(I) species was investigated using resonance Raman spectroscopy at 643 nm. At this excitation wavelength only the catalytically competent intermediate, i.e., the Rh(I) complex is absorbing. While we observed the Raman band of the carbonyl function at 1694 cm^{-1} for the initial Rh(I) species, indicating an intact ester functional group at the bipyridine chromophore, this band vanishes after one or more exposure steps to low pH values. This is most likely because the functional group is hydrolyzed. While the C=O double bond is initially detected with an intact ester functional group, when the ethoxy group is cleaved at lower pH, the charges become delocalized, and the carbonyl vibration disappears. In addition, since the π electrons of the carbonyl group are also conjugated to the bipyridine chromophore, the change in the ester functional group also impacts the intensity ratios of the bipyridine bands, as can be seen when comparing the spectra of Rh(I)-0 (i.e. **HEC3** exposed to no pH swing) with Rh(I)-1 to -4 (i.e., **HEC3** exposed to 1 to 4 pH swings; see Fehler! Verweisquelle konnte nicht gefunden werden.)”*

Fig. S17 Experimental resonance Raman spectra of chemically reduced **HEC3** in aqueous solution after exposure to n acid/base pH swings ($n = 0-4$), excited at 643 nm.

The corresponding conclusions that were drawn from this set of investigations was included into the main manuscript at the end of chapter 2.4. and reads as follows (pages 15-16):

*“For **HEC3**, this pH-induced degradation was confirmed by performing resonance Raman measurements of the catalyst exposed to the conditions of the acid base cycle during catalytic charging and discharging (see **Fig. S17**). It could be shown that upon addition of acid, already at the first discharging step the ester functionality at the bipyridine was cleaved. From cycle 2 onwards, H_2 evolution is thus no longer catalyzed by **HEC3** but by a structurally altered bipyridine-based Rh complex.”*

Point-by-point response to the reviewers' comments

Reviewer 1:

In the revised version of the manuscript, the authors have provided a quite thorough reply to my prior comment requesting that the fate of the molecular catalyst be examined. Through very detailed analysis involving several techniques, the authors have identified a degradation pathway of their catalyst involving hydrolysis of ester groups under low-pH conditions. I emphasize here that the identification of this degradation pathway in no way tampers my enthusiasm for this report--rather, the very thorough and chemically sensible conclusions drawn by the authors add significantly to the quality of the science presented here. It is a lamentable draw-back of many chemical studies of molecular catalysts for energy conversion and storage that degradation pathways and longevity studies are not studied in more detail--this study now overcomes this common problem. I have reviewed the new data from Raman studies provided by the authors and agree with their interpretations.

Response: We thank the reviewer for his suggestion to elucidate the fate of the catalyst by suitable methods. We agree that such studies are rarely reported but would add significant benefit to many reports.

In terms of the path to wrap this up, I have only a single final suggestion for their consideration. On page 2 of their response document, they refer to a d8 to pi* MLCT for the Rh(I) species. The noted spectral features between 540 and 840 nm, from what I know of species of this type, given them an appealing dark coloration (in derivatives that I have studied, purple). However, rather than just being a d8 to pi* transition, this coloration can be also thought of arising from mixed [Rh(I) d8 + bpy pi*] to [pi*] transitions, owing to delocalization of charge at the Rh(I) state into the first LUMO of the bpy system. The evidence for this comes often from vibronic features that, granted, are not obvious in the new spectra in Fig S16 and shown on p. 3 of the response document. So, considering all this, the authors may wish to adjust their interpretation/description of the origins of

the new beautiful absorptions slightly. This is optional however and I emphasize that this does not need to be reviewed by me again.

Response: We thank the reviewer for pointing out this issue. Indeed, the absorption band of the reduced HEC3 (now termed compound 3 according to the editorial requests) between 540 and 840 nm is obtained from a superposition of excited states involving Rh(I) $d8 \rightarrow$ bpy MLCT AND bpy $\pi^* \rightarrow \pi^*$ (partially reduced bpy-ligand) transitions, as known from the literature (DOI: 10.1039/B504286K). Nevertheless, the expected vibrationally structured absorption bands from the bpy radical anion in the visible and near-infrared regions is difficult to detect as these bands are superimposed by the absorption of the Rh(I) \rightarrow bpy MLCT.

A shortened version of this response has been included on page 8 of the SI document.

This is a lovely contribution to the literature of Rh catalysis, energy conversion, and the opportunities and challenges of building systems for energy storage in chemical bonds. I congratulate the authors on their exceptionally beautiful and thorough work. And, I look forward to seeing it in print for all to read very soon.